# A deep learning framework to predict binding preference of RNA constituents on protein surface

Jordy Homing Lam[1,2,13], Yu Li [1,13], Lizhe Zhu [2,3]*, Ramzan Umarov[1], Hanlun Jiang[4], Amélie Héliou[5], Fu Kit Sheong[2], Tianyun Liu[6], Yongkang Long [1,7], Yunfei Li[7], Liang Fang[7], Russ B. Altman[6], Wei Chen[7]*, Xuhui Huang [2,8,9,10,11,12]* & Xin Gao [1]*

Protein-RNA interaction plays important roles in post-transcriptional regulation. However, the task of predicting these interactions given a protein structure is difficult. Here we show that, by leveraging a deep learning model NucleicNet, attributes such as binding preference of RNA backbone constituents and different bases can be predicted from local physicochemical characteristics of protein structure surface. On a diverse set of challenging RNA-binding proteins, including Fem-3-binding-factor 2, Argonaute 2 and Ribonuclease III, NucleicNet can accurately recover interaction modes discovered by structural biology experiments. Furthermore, we show that, without seeing any in vitro or in vivo assay data, NucleicNet can still achieve consistency with experiments, including RNAcompete, Immunoprecipitation Assay, and siRNA Knockdown Benchmark. NucleicNet can thus serve to provide quantitative fitness of RNA sequences for given binding pockets or to predict potential binding pockets and binding RNAs for previously unknown RNA binding proteins.

[1] Computational Bioscience Research Center, Computer, Electrical and Mathematical Sciences and Engineering Division, King Abdullah University of Science and Technology (KAUST), Thuwal 23955-6900, Saudi Arabia. [2] Department of Chemistry, The Hong Kong University of Science and Technology, Hong Kong, China. [3] Warshel Institute for Computational Biology, School of Life and Health Sciences, the Chinese University of Hong Kong (Shenzhen), Shenzhen, 518172 Guangdong, China. [4] Department of Biochemistry and Institute for Protein Design, University of Washington, Seattle, USA. [5] Laboratoire d' Informatique, Department of Computer Science, École Polytechnique, Palaiseau, France. [6] Departments of Medicine, Genetics and Bioengineering, Stanford University, Stanford, CA, USA. [7] Department of Biology, Southern University of Science and Technology, 518055 Shenzhen, Guangdong, China. [8] Division of Biomedical Engineering, The Hong Kong University of Science and Technology, Hong Kong, China. [9] State Key Laboratory of Molecular Neuroscience, The Hong Kong University of Science and Technology, Hong Kong, China. [10] Hong Kong Branch of Chinese National Engineering Research Center for Tissue Restoration & Reconstruction, The Hong Kong University of Science and Technology, Hong Kong, China. [11] Institute for Advanced Study, The Hong Kong University of Science and Technology, Hong Kong, China. [12] HKUST-Shenzhen Research Institute, Hi-Tech Park, 518057 Nanshan, Shenzhen, China. [13]These authors contributed equally: Jordy Homing Lam, Yu Li *email: zhulizhe@cuhk.edu.cn; chenw@sustech.edu.cn; xuhuihuang@ust.hk; xin.gao@kaust.edu.sa

After transcription, mRNAs undergo a series of intertwining processes before being finally translated into functional proteins. These post-transcriptional regulations, which provide cells an extended option to fine-tune their proteomes, are in general mediated through interactions between RNAs and RNA-binding proteins (RBPs). In cells, RNAs are largely regulated by two modes of specific interactions – either by direct recognition of RNA motifs on the RBP surface or by an indirect RNA-guided manner. In the former case, the RBP makes direct contact with the bases of RNA. For instance, the Pumilio/FBF (PUF) family can control translations via direct base-protein contact, e.g., with UGUR motifs on RNA transcripts[1]. In the latter case, the RBP interacts with backbone or non-Watson-Crick (WC) edges of the bases leaving WC-edges for target recognition. For example, in core enzymes of RNA interference (RNAi, e.g., Argonautes) and gene-editing complexes (e.g., CRISPR-Cas), selective loading of a guide-RNA (gRNA) into the RBP is a prerequisite to activate the enzyme; target D/RNA recognition is then mediated through the WC edges of gRNA while other parts of the gRNA remain in contact with the RBP. Therefore, deciphering the specificity and mechanisms in RNA-protein interactions is of fundamental importance to understanding the functions of RBPs, identifying RBPs, and designing RNAs for RBP recognition and regulation.

To approach systematic mapping of these interactions, various experimental and computational techniques have been developed. In the experimental genre, in vivo UV-crosslinking immunoprecipitation assays such as CLIP-HITS[2] and in vitro selection assays such as HT-SELEX[3] and RNAcompete[4] are among the most successful technologies. In general, specificity patterns obtained from these methods can be expressed as the logo diagram for each RBP or as analytical scores for individual RNA sequences. Through structure elucidation techniques, binding mechanisms for many of these characterized RBPs, e.g., hnRNP, Nova and PAZ, have also been clarified[5–7]. However, despite such remarkable achievements, experimental assays are constrained by reactivity, detection, and scalability limits. For instance, UV-crosslinking assays prefer uridine-rich sequences, because pyrimidines are more photoactivatable than purines[8]. Although arguably the chemical origin of these assayed specificities can be validated by ribonucleoprotein co-crystals, single or a few such co-crystals could hardly explain the genuinely ambiguous patterns on logo diagrams (e.g., specific to both U and A on the same position).

To this end, computational approaches can enhance experimental results. In this genre, the body of sampled experimental knowledge, assays, and structures, can be refined to uncover previously mis-/un-acknowledged specificity patterns. Exemplary assay-based computational approaches, e.g., DeepBind and variants[9], can integrate and learn over assay data collected for an RBP to infer the specificity pattern that is consistent with large-scale assays. There are also less explored structure-[10] and sequence-based[11] computational approaches. Typically, in these latter approaches, given a three-dimensional protein structure or its amino acid sequence, local protein sequence context among other structural information (e.g., solvent accessibility, secondary structure, hydrophobicity, and electrostatic patches) can be extracted in units of residues and used to train models in reference to RNA-RBP structures in the Protein Data Bank (PDB). As such, the demand for experimental data to start with is relaxed from assay-based methods. However, due to the highly limited amount of available features, their predictive power is restricted to distinction of RNA-binding sites from non-sites, i.e., binary predictions made over locations or indices of protein residues without suggesting the preferred base/sequence nor any informative interaction modes (e.g., via backbone or base).

Nevertheless, computational approaches are scalable and cost-efficient, thus are important complements to experimental techniques.

In this work, we introduce NucleicNet, a structure-based computational framework, which addresses topical challenges presented above: (i) we developed ways to learn efficiently from the PDB such that we can predict interaction modes for different RNA constituents – Phosphate (P), Ribose (R), Adenine (A), Guanine (G), Cytosine (C), Uracil (U), and non-site – and visualize them on any protein surface; (ii) NucleicNet requires no external assay input to derive logo diagrams consistent with assay data, including RNAcompete, Immunoprecipitation Assay, and siRNA Knockdown Benchmark; (iii) the logo diagrams or position weight matrices (PWMs) obtained from NucleicNet can be used to score the binding potential of individual RNA sequences; (iv) NucleicNet can generalize across different families of RBPs and be potentially used to identify new RBPs and their binding pockets/preferences. Our pipeline is founded upon the FEATURE vector framework[12], which encodes physicochemical properties on protein surfaces as high-dimensional feature vectors. This rich vector space not only has covered most features developed in other programs, but can also account for subtle differences in local topologies via its discrete radial distribution setup. Importantly, learning from these high-dimensional feature space is nontrivial, therefore a deep residual network is proposed and trained for this purpose.

We benchmark NucleicNet from three different data sources – structural, in vitro, and in vivo experiments. For structural data, two tests were done: (i) in reference to an external benchmark[11], we show that NucleicNet can effectively outperform all available sequence-based methods in differentiating RNA-binding sites and non-sites on protein surfaces; and (ii) when compared to our own carefully constructed non-redundant 7-class dataset, we show that NucleicNet can resolve RNA constituents with a class-averaged AUROC of 0.77 with respect to all 6 RNA constituents and non-sites, and of 0.66 with respect to the 4 bases. For in vitro data, the RNAcompete (RNAC) assay is adopted to assess the accuracy of our NucleicNet PWMs in dealing with RBPs that directly recognize RNA motifs on their surfaces. In all eight available examples, we show that, without any training on the assay data, NucleicNet PWMs are comparable to RNAC PWMs in identifying best binding 7-mers from all possible 7-mer sequences. Finally, we also explored downstream applications relevant to in vivo RNAi experiments. We show that the NucleicNet score is capable of explaining in vivo asymmetry in the guide strand loading of human Argonaute 2 (hAgo2) as well as the varied knockdown levels in different siRNA sequences.

## Results

**An overview of NucleicNet.** In NucleicNet, our goal is to predict on each location (grid point) of a protein's surface, whether the physicochemical environment presented on-site is fit to bind with an RNA and, if affirmative, the binding preference to each type of RNA constituent – Phosphate (P), Ribose (R), Adenine (A), Guanine (G), Cytosine (C), and Uracil (U) – that binds to the location. Computationally, we cast the problem as a supervised seven-class classification problem. Accordingly, we formulate the end-to-end training of NucleicNet as follows (Fig. 1 top panel). First, surface locations on ribonucleoprotein complexes are retrieved from the PDB and typified as 7 classes that correspond to the bound RNA constituents and non RNA-binding site (X). Corresponding physicochemical environment on each location is then characterized using the FEATURE program[12] (Methods, Fig. 1 middle panel). Next, a deep residual network is trained to associate each physicochemical environment with one of the 7

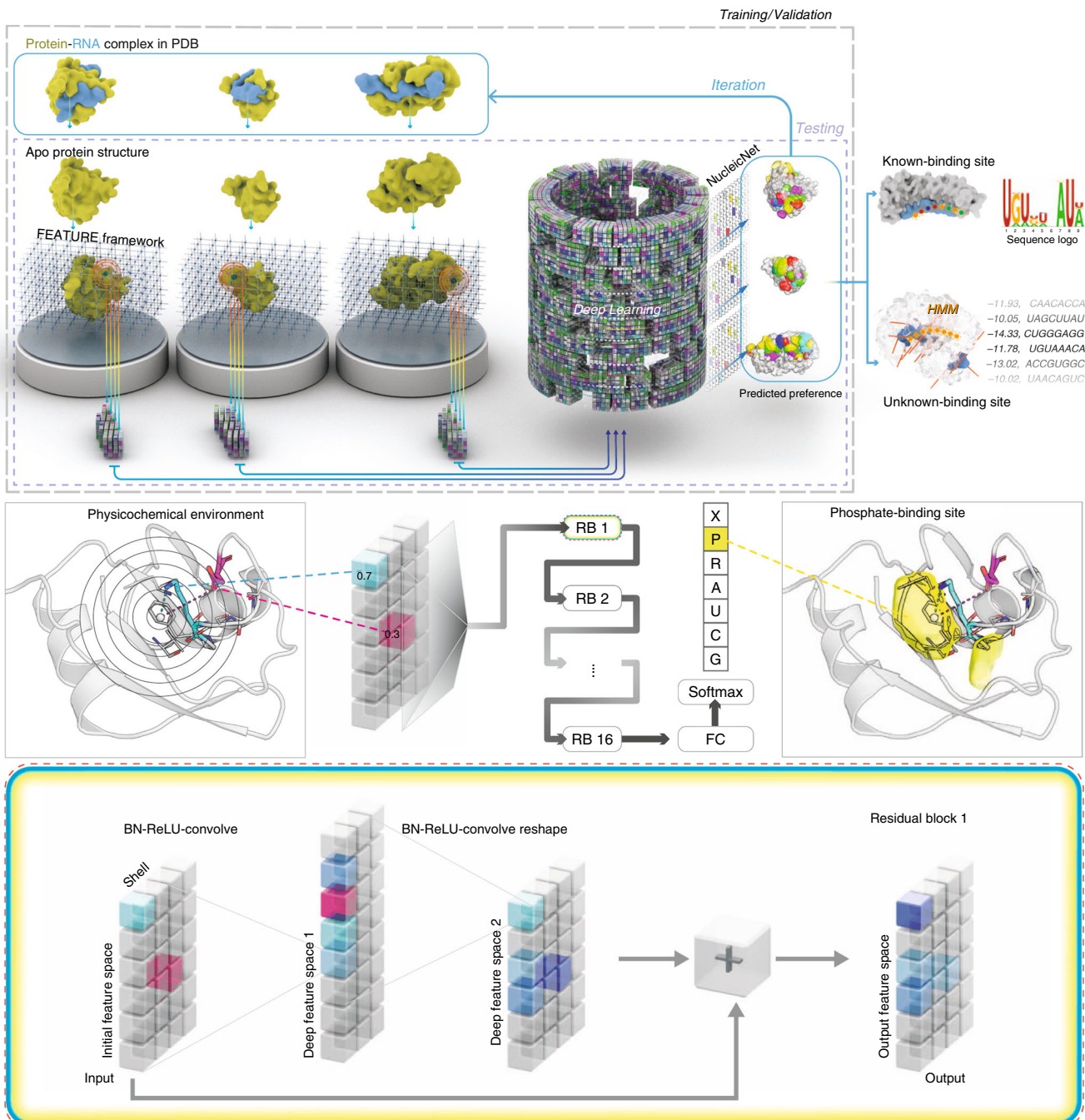

**Fig. 1** Overview of NucleicNet. Top panel: training strategy and utilities of NucleicNet. Ribonucleoprotein structures in the PDB are stripped of their bound RNA. Grid points are placed on the surfaces of the proteins. Each of these grid points (locations) is analyzed by the FEATURE program for their surrounding physiochemical environment, which is encoded as a feature vector. Binding locations for all six classes of RNA constituents and non RNA-binding sites are labeled accordingly. The labels together with their respective vectors are compiled; this formulates the training input for a deep residual network. Parameters in the network are iteratively updated by backpropagation of errors and are trained to differentiate the seven classes. Once training is complete, the learned model can then be used to predict binding sites of RNA constituents for any query protein structure surface. Downstream applications of the prediction outcome includes production of logo diagrams for RBPs and a scoring interface for any query RNA sequence. Middle panel: physicochemical environment accession and introduction on the residual network. In the FEATURE vector framework, physicochemical environment is perceived by accounting properties on atoms of a protein within 7.5 Å of a grid point in a radial distribution setup. As such, space surrounding each grid point is divided into six concentric shells of spheres and, for each of these shells, 80 physicochemical properties (e.g., negative/positive charges, partial charges, atom types, residue types, secondary structure of the possessing residue, hydrophobicity, solvent accessibility, etc.) were accounted resulting in a tensor of dimension 6×80. The tensor is then transformed by a deep residual network with 16 sequential residual blocks. After that, the final residual block is connected to a fully connected layer with a softmax operation to assess binding probability of each class on that location. Bottom panel: we illustrate the principle operations in a residual network, namely batch normalization (BN), rectified linear unit (ReLU), locally connected network, and the quintessential skip connection, which adds the initial input back to the penultimate output layer

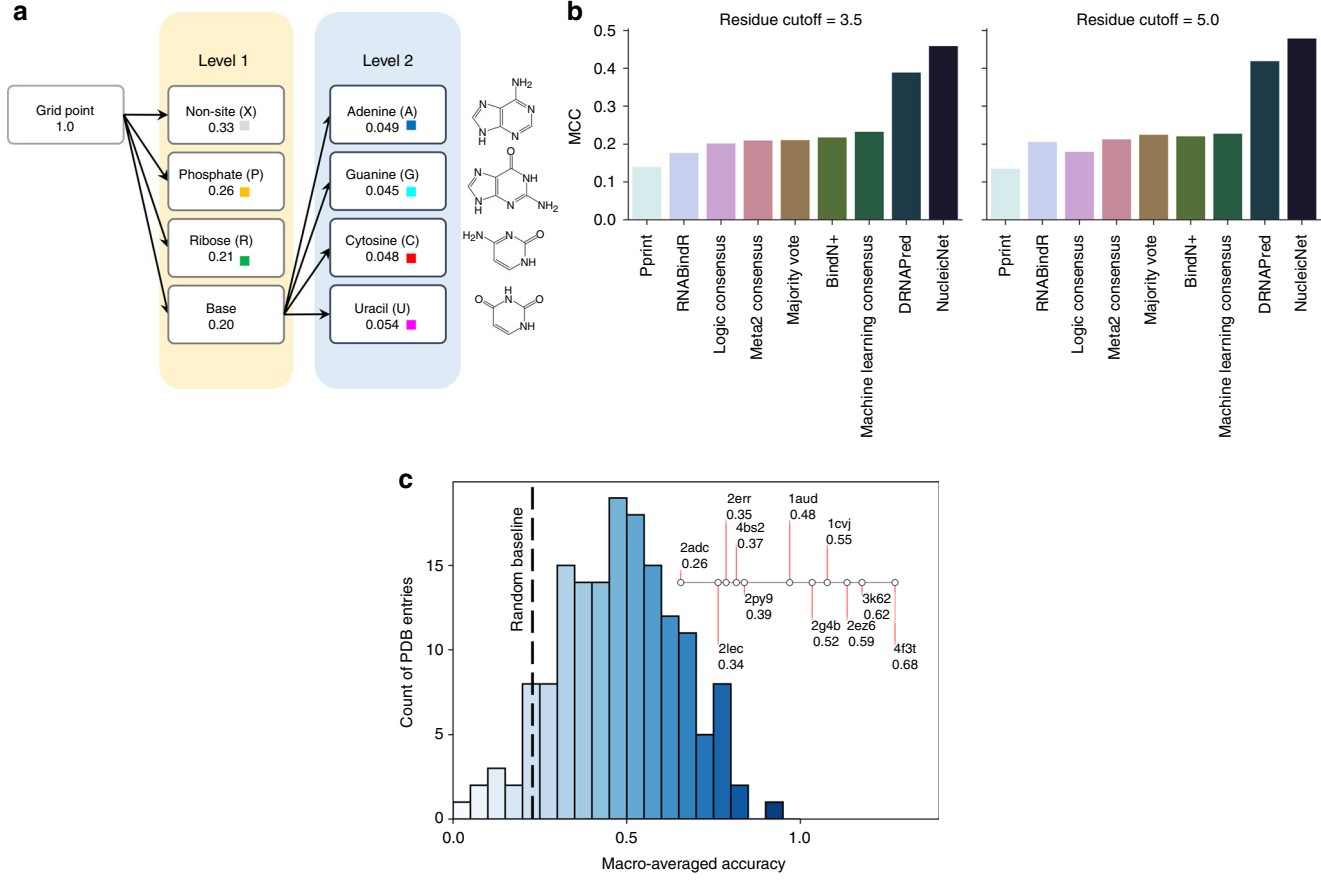

**Fig. 2** Data statistics and performance of NucleicNet. **a** Statistics and hierarchy of a 7-class classification. Ratio of data available for each class is shown in the second line of each box. Each grid point is first classified under 4 coarse labels – non-site, phosphate, ribose, and base – and then 4 fine base labels – Adenine(A)/Guanine(G)/Cytosine(C)/Uracil(U). Color code of each constituent is shown in as a square. **b** Benchmark of NucleicNet to distinguish sites from non-sites. All methods[13-16] listed from the recent review paper[11] were compared using Matthew's Correlation Coefficient (MCC) in terms of the two cutoffs, in angstrom, as indicated in title. **c** Benchmark of NucleicNet to distinguish among the six RNA constituents and non-sites in 3-fold cross validation of the protein-RNA complex structures in PDB. A histogram of macro-averaged accuracy is provided. Baseline accuracy (0.23) referring to a random 7-class predictor is indicated with the dash line

classes in a hierarchical manner (Methods, Fig. 2a). Finally, parameters of the network are optimized through standard backpropagation of the categorical cross entropy loss. Note that training data are entirely derived from three-dimensional structures in the PDB, i.e., we used no training data from external assays. Once training is completed for NucleicNet, raw surface characteristics extracted with FEATURE on the surface location of the query protein can then be fed forward to infer binding preference for each class on a location-by-location basis.

One strength that distinguishes our approach from related work is that not only binding sites of all 6 classes of RNA constituents are predicted and visualized on the surface of protein, but also, at the same time, these detailed results can be assimilated into logo diagrams or scoring interface for RNA sequences. As such, outcomes from the feed-forward module are packaged into three utility modules – a Visualization module that indicates top predicted RNA constituents as a surface plot (Fig. 3a–c), a Logo Diagram module that generates the logo diagram when the RNA binding pocket on the protein surface is known (Fig. 4a–h), and a Scoring Interface module to apprehend binding score for a query RNA sequence (Figs. 4a–h and 5a, b), which can predict the most likely RNA sequence and the corresponding binding pocket on any query protein (Fig. 3a–c). The latter two modules can be summarized as a hidden Markov model (HMM), which incorporates both the locations of the

bases and the geometric constraints for feasible RNA sequences (Methods, Supplementary Figs. 4 and 5). The Visualization module is used to compare our predictions with structural biology experiments. The Logo Diagram and Scoring modules are used to compare our predictions with in vivo or in vitro assay data.

**Validation of NucleicNet from structural perspectives.** Various reliable ground truths can be extracted from known ribonucleoproteins structures deposited in the PDB. First, we start with distinguishing RNA-binding residues from non-RNA-binding ones, i.e., a binary classification. This is a classical problem tackled by most computational predictors on protein-RNA interaction[11]. In general, a protein residue is considered RNA-binding in a co-crystal if at least one of its atoms is within a certain distance from atoms of the RNAs. In a recent review[11], both 3.5 Å and 5.0 Å cutoffs were considered. The benchmark RNA_T dataset proposed therein[11], which consists of 175 RNA-binding protein chains, was generated by clustering protein chains with respect to their sequence and structural similarities, where annotations of RNA-binding residues were transferred among similar chains to alleviate effects of strand truncations[11]. Based on this ground truth, we benchmarked NucleicNet with a broad range of state-of-the-art predictors based on sequence information (Fig. 2b). To

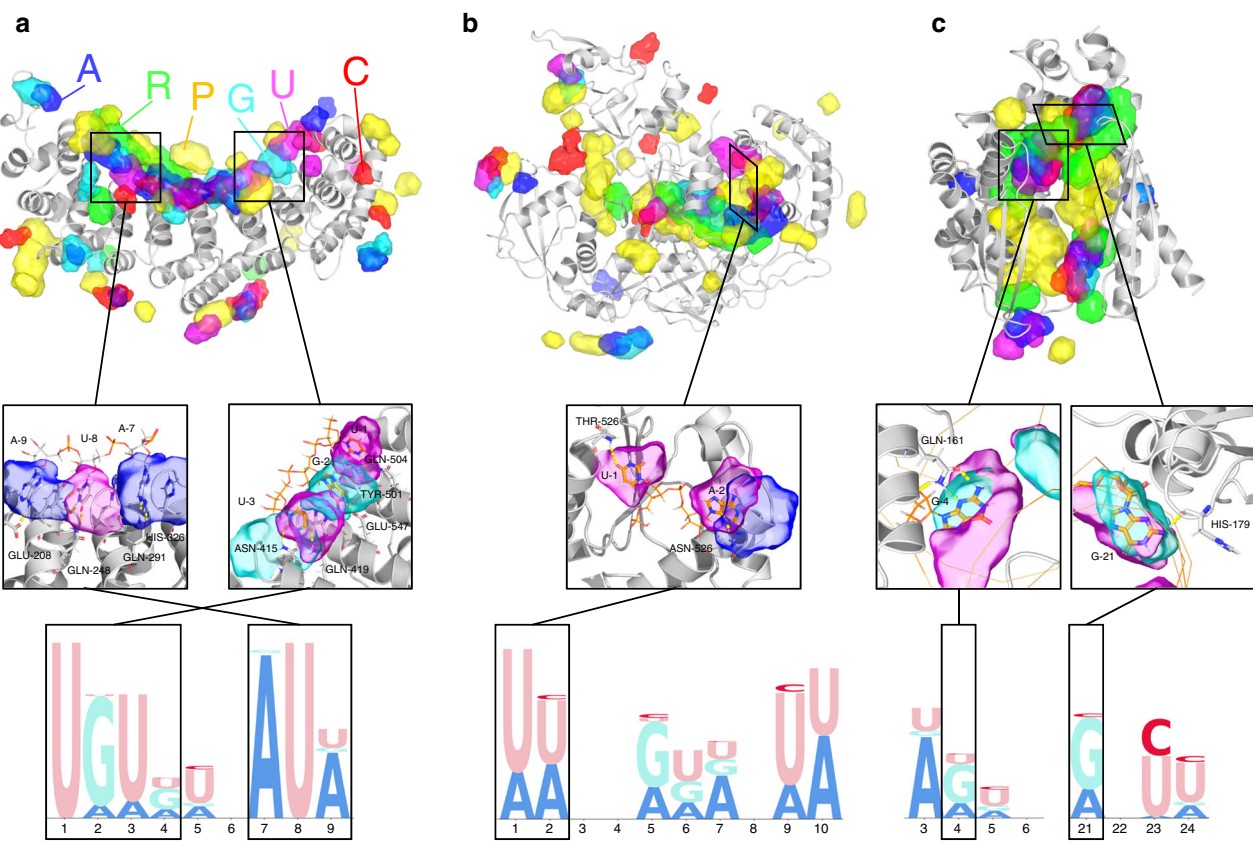

**Fig. 3** NucleicNet prediction captures detailed binding motion determined by structural biology experiments. **a** FBF2, **b** hAgo2, and **c** Aa-RNase III. Upper panel: NucleicNet predictions for query RBPs. Top 10% of the binding sites are drawn in transparent surfaces for each class. Color code: phosphate-yellow, ribose-green, adenine-blue, cytosine-red, guanine-cyan, uracil-purple, and protein-gray cartoon. Middle panel: detailed view on chemical interactions. Lower panel: predicted sequence logo diagrams for respective RBPs

assign a binary label (site or non-site) on each protein residue using our NucleicNet predictor that works on grid points over the protein surface, score vectors on 30 grid points closest to a protein residue were taken to vote for 2 coarse classes, namely 'RNA-binding site' and 'non-site'; the 6 finer classes (that correspond to individual RNA constituents) are considered 'RNA-binding site'. Testing benchmark proteins[11,13] are omitted from training. At both aforementioned ranges of distance cutoffs, NucleicNet outperforms all available methods[13–16] (Fig. 2b). This therefore demonstrates the basic utility of NucleicNet as a tool to predict general RNA-binding sites.

Next, we evaluate on NucleicNet's ability to retrieve binding sites for the six detailed RNA constituents proposed; this includes Phosphate (P), Ribose (R), Adenine (A), Guanine (G), Cytosine (C) and Uracil (U). A 3-fold cross-validation was performed over a carefully selected and curated non-redundant dataset from all protein-RNA complex structures from PDB (see Methods), which consists of 158 complex structures, resulting in about 280,000 grid points in the dataset. We divided the 158 proteins into three folds. Each time, two folds of them were used for training and one fold for testing. Between folds, BLASTClust sequence homology of ≥90% was disallowed (see Methods). Notice that the granularity of this cross-validation is individual proteins, instead of grid points, which eliminates bias in the size of proteins. Table 1 reports the performance in terms of AUROC, F1-score, Precision and Recall for each class (Metrics explained in SI). For the bases (A/U/C/G), an AUROC of 0.66 can be achieved in average. Remarkably, the power to differentiate sites and non-sites is recapitulated in an AUROC of 0.97. Categorical accuracy with respect to each protein is also calculated. A distribution of

the accuracy score is shown in Fig. 2c; proteins covered in case studies (Figs. 3a–c and 4a–h) are marked out with their PDBID on the inset line diagram to indicate their performance, which shows that the accuracy of the case studies spreads over a wide range. In general, a median accuracy of 49% is achieved in the non-redundant 3-fold cross validation (c.f. random baseline 23%, Supplementary Note 1). This proof-of-principle analysis therefore demonstrates that NucleicNet can learn from a diverse structural database of physicochemical environment and generalize to unseen RBPs to recall potential binding RNA constituents, provided that structure of the elucidated protein is largely intact and contains relevant RNA-binding domains.

**Complex spatial patterns of RNA-binding sites reproduced by NucleicNet.** One strength that structure-based methods offer is their potential to reveal and visualize binding sites on protein surfaces. While previous structure-based methods concern only binary classifications (sites and non-sites), our method can illustrate further on all six common RNA constituents – 'Phosphate' (P), 'Ribose' (R), 'Adenine' (A), 'Guanine' (G), 'Cytosine' (C), and 'Uracil' (U). We demonstrate this unique power of our method via three exemplary RBPs: Fem-3-binding-factor 2 (FBF2, PDB Entry 3k62, Fig. 3a), Human Argonaute 2 (hAgo2, PDB Entry 4f3t, Fig. 3b), and Aquifex aeolicus Ribonuclease III (Aa-RNase III, PDB Entry 2ez6, Fig. 3c). FBF2 is an example from RBPs that interact directly with single-stranded RNA (ssRNA) motifs through base contacts, while the hAgo2 is an example from RBPs that functions in an RNA-guided manner through backbone or non-WC edge contacts. The third example, Aa-RNase III, involves double-stranded RNA-binding domain

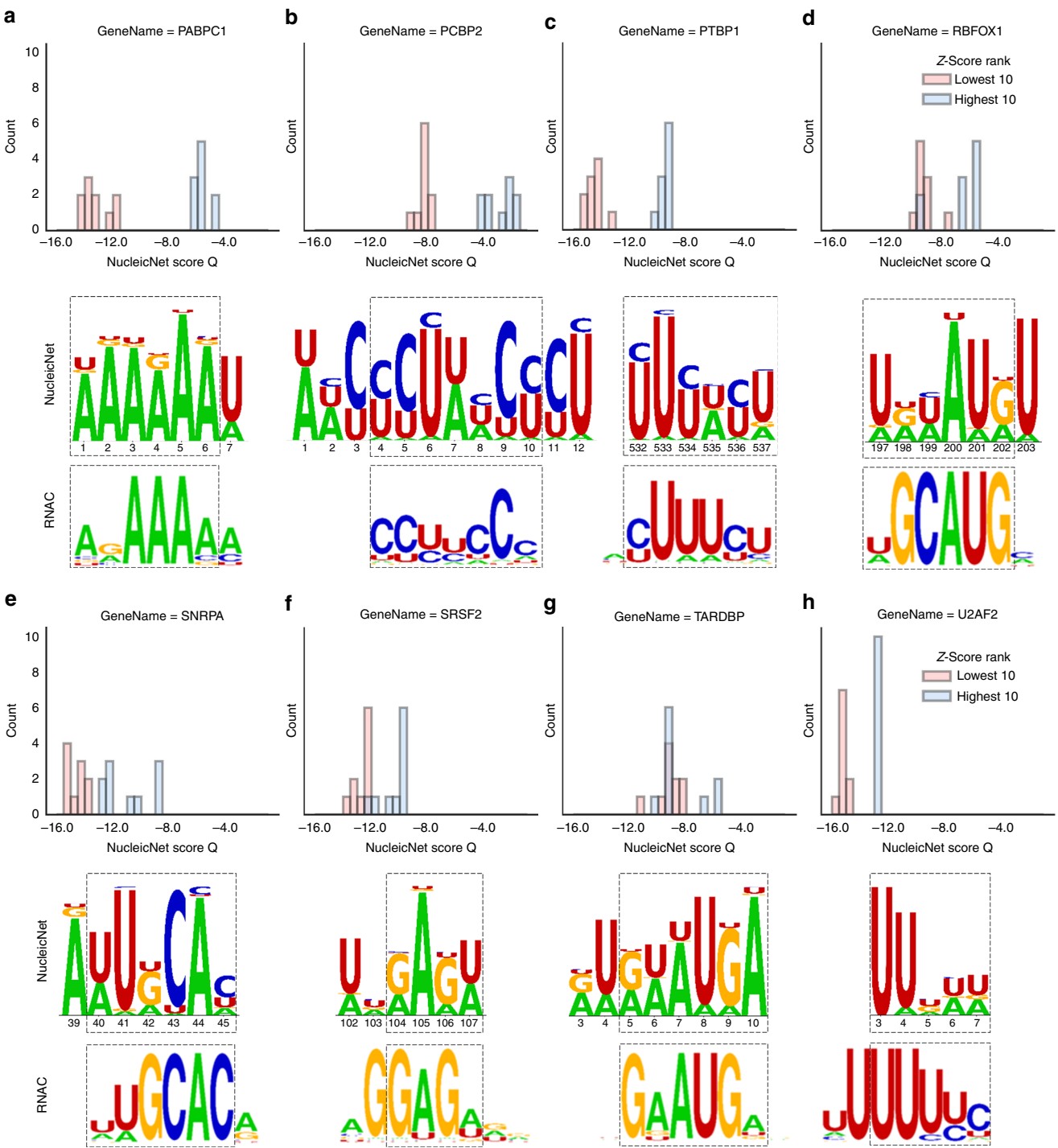

**Fig. 4** Comparing scores and logo diagrams of NucleicNet with those obtained from RNAcompete (RNAC) assay. **a** PABPC1, **b** PCBP2, **c** PTBP1, **d** RBFOX1, **e** SNRPA, **f** SRSF2, **g** TARDBP, and **h** U2AF2. Upper panel: we show that the NucleicNet score is capable of differentiating top and bottom 10 sequences indicated by the RNAC Z-score. Lower panel: we compare sequence logos generated by NucleicNet with that by RNAC. Best matching sections of letters are highlighted with a dash box

(dsRBD). In Fig. 3, we indicate top predicted binding sites on these proteins for each binding class using our visualization module. In all cases, predictions were made on the protein structure after removing RNAs from the ribonucleoprotein complex. These proteins and their homologs were all excluded from the training process. In Fig. 3 middle panel, we show that strong preference for nucleobases are mostly found at places where nucleotides interact explicitly with protein residues when superposed on a ribonucleoprotein structure. In Fig. 3 lower panel, sequence logo diagrams were generated by averaging the NucleicNet score at the nucleobase locations on the long native RNA strand (Methods). In all cases, we show that NucleicNet has reproduced the detailed binding specificity captured by structural biology experiments.

**Fem-3-binding-factor 2.** The PUMILIO/Fem-3-binding-factor (PUF) family of RBPs are important post-transcriptional

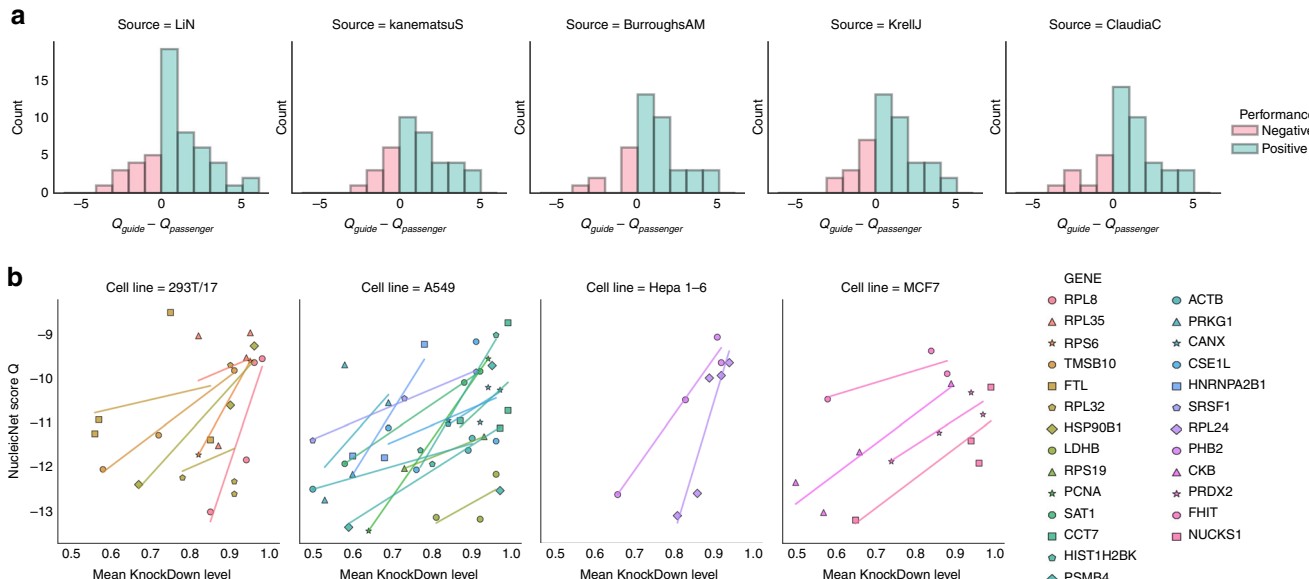

**Fig. 5** NucleicNet predictions agree with in vivo experiments on Ago2. **a** Histogram of the difference in the NucleicNet score between the guide and the passenger strands from five different cell lines, namely four from human (BurroughsAM: acute monocytic leukemia THP-1[26], KanematsuS: colon cancer DLD[27], KrellJ: colon cancer HCT116[28] and ClaudiaC: T cell leukemia[29]) and one from mouse (LiN: neuroblastoma N2a[30]). Bars with positive difference values are highlighted in green else if negative in red. **b** Comparing the NucleicNet score with benchmark mean knockdown levels of shRNA targeting 26 genes in four different cell lines

**Table 1 Statistics of performance in cross validation of the non-redundant dataset from PDB**

| Metrics | NonSite | Phosphate | Ribose | Adenine | Guanine | Uracil | Cytosine |
|---|---|---|---|---|---|---|---|
| AUROC | 0.97 | 0.93 | 0.84 | 0.67 | 0.67 | 0.65 | 0.66 |
| F1-score (macro) | 0.90 | 0.70 | 0.63 | 0.47 | 0.38 | 0.48 | 0.32 |
| Recall (macro) | 0.88 | 0.82 | 0.63 | 0.46 | 0.38 | 0.45 | 0.37 |
| Precision (macro) | 0.92 | 0.61 | 0.63 | 0.48 | 0.38 | 0.51 | 0.29 |
| F1-score (micro) | 0.90 | 0.70 | 0.64 | 0.47 | 0.41 | 0.48 | 0.32 |
| Recall (micro) | 0.88 | 0.81 | 0.64 | 0.47 | 0.40 | 0.46 | 0.36 |
| Precision (micro) | 0.92 | 0.61 | 0.64 | 0.48 | 0.41 | 0.51 | 0.29 |

regulators. In a typical PUF–mRNA interaction, the PUM-HD domain, common among all PUFs, will bind to the 3′ untranslated region of mRNA that contains a conserved UGUR sequence motif. The strong sequence specificity is mediated through direct interactions (aromatic stacking and hydrogen bonds) made between protein surface residues and RNA nucleobases[1]. The Fem-3-binding-factor 2 (FBF2) is one of the best-characterized PUF family proteins. In Fig. 3a middle panel, we show that interacting surface indicated by NucleicNet at Q504/Q419/N415/E542/Y501 and Q248/Q291/E208/H326 of FBF2 largely involves hydrogen bond donors or acceptors on the PUM-HD repeats. The respective sequence logo diagram derived from these locations (Fig. 3a lower panel) indicates a strong sequence preference at base 1–4 and 7–8 that is consistent with the 5′-UGUR and downstream A7–U8 pattern reported previously[1,17]. In addition, NucleicNet also correctly captures the modest preference for A or U (A > U > G) at base 9 consistent with the consensus reported by yeast three-hybrid assays[17,18], even though the crystal-bound native base at that position is a C. This therefore suggests that NucleicNet is able to reveal underlying sequence specificity patterns unseen in crystal structures and in the absence of third-party assay data.

**Human Argonaute 2.** Human Argonaute 2 (hAgo2) is an exemplary RBP that operates in an RNA-guided manner, where the guiding RNA strand can be a small interfering RNA (siRNA) or a micro RNA (miRNA). In cells, both of these RNAs pre-exist as a duplex of complementary single-strands. However, during assembly of the RNA-induced silencing complex (RISC), often one of the strands is preferentially loaded into hAgo2 to guide cleavage of the target RNAs. This asymmetric behavior is heavily affected by small changes in RNA sequences of the precursor duplex[19]. Two attributing factors were identified – (1) weakening of the base pair at one of the 5′ ends, this decides which strand will unwind at its 5′ end and subsequently enter the RISC complex[19]; (2) guiding RNA-hAgo2 interactions at base 1 (Fig. 3b middle panel) and the non-Watson-Crick edges of base 2–8 (the seed region) (Fig. 3b), these interactions are hypothesized to lower the enthalpic cost of RISC assembly[20–22]. However, compared to intensive studies on target RNA recognition by the RISC complex, the second factor, concerning RISC assembly that correlates loading and knockdown efficiency with guiding RNA-protein interaction, is much less explored.

In Fig. 3b upper panel, we show that binding sites of the guide strand, including the phosphate-ribose backbone around PAZ and N domains, are correctly captured by NucleicNet. Specifically, in Fig. 3b middle and lower panels, we focus on the 5′-end binding pocket on the Mid domain and show that NucleicNet correctly predicts a strong U-binding pocket (U > A ≫ C/G) at base 1 and a U/A binding pocket (U = A) at base 2. The first preference on base 1 and its order are well supported by structural

evidence and NMR titration experiments performed using nucleoside monophosphates (mimics of the 5′ end, UMP (0.12 mM) > AMP (0.26 mM) ≫ CMP (3.6 mM)/GMP (3.3 mM))[20]. For other binding preferences in the seed region, only structural evidence is available and it is scattered among different PDB entries containing different seed sequences. For example, in the PDB entry 4f3t, A2 and G5 interact with N562 and Q757 respectively; in PDB entries 5js1/5t7b, U2 interacts with N562. These results are consistent with the logo diagram provided by NucleicNet (Fig. 3b lower panel). We show later in Fig. 5a, b that these NucleicNet predictions are supported by immunoprecipitation experiments and knockdown assays affirming that guide loading efficiency and sequence-protein interactions are correlated.

**Aquifex aeolicus Ribonuclease III.** Double-stranded RNA-binding domain (dsRBD) is a domain that widely occurs among double-stranded RNA-specific endoribonucleases, including the Aa-RNase III presented here. Originally, recognition of RNAs in dsRBDs were thought to be shape-dependent rather than sequence-specific. However, recent structural evidence confirms that this domain can recognize bases by interacting with the minor groove[23]. In Fig. 3c middle and lower panels, we show that NucleicNet has correctly predicted two strong G-binding sites concentrated around H179 and Q161, corresponding to the first α helix and the loop between β strands 1 and 2 of the dsRBD, which agree well with the existing co-crystals.

**Validation with in vitro RNAcompete assay data.** To validate NucleicNet on RBPs that directly recognize RNA motifs on its surface, we compare the NucleicNet score with scores obtained from the RNAcompete assay (RNAC)[24,25]. RNAC is a large-scale in vitro experiment that uses the epitope-tagged RBP to competitively select RNA sequences from a designed pool. For each RBP, 7-mer RNA-binding profiles obtained can be summarized as a Z-score for the individual RNA sequence or as a PWM by aligning the top 10 scoring sequences. Higher Z-score indicates better binding. We tested NucleicNet on all the RBPs for which both RNAC data and PDB structures are available (PABPC1, PCBP2, PTBP1, RBFOX1, SNRPA, SRSF2, TARDBP, and U2AF2). In all cases (Fig. 4a–h, Table 2), a Welch's t-test is performed and shows that NucleicNet is capable of differentiating between the top and bottom 10 sequences indicated by RNAC Z-scores with a positive test-statistics and p-value < 0.005 except for TARDBP, where its RNAC binding profile is specific to a single sequence. In Supplementary Table 2, we further compare the NucleicNet score with the RNAC PWM score in different rank ranges of RNAC Z-scores (top/bottom 10, 50, and 100). In all cases, NucleicNet is capable of differentiating the sequences, although it was never trained on any assay data. This therefore suggests that the NucleicNet score is predictive and is suitable to complement selection assays.

Interestingly, NucleicNet is able to predict binding preference that is beyond structural biology information in PDB. For example, all the three PDB entries for protein PTBP1 (PDBID: 2adc, 2adc, and 2ad9) are bound with the RNA sequence CUCUCU, which deviates from the RNAC suggested sequence YUUUYU (Table 2). This suggests that single or few PDB co-crystal structures may not inform about RNA-binding preference comprehensively. However, by integrating with other PDB data through training, NucleicNet predicts a suggested sequence of UUUWYU in reasonable agreement with the RNAC sequence (Fig. 4c), which indicates its ability to make predictions that are not present in the training data. Accordingly in these cases, NucleicNet can have low accuracy scores with respect to PDB

**Table 2 Statistics of performance and suggested sequences from NucleicNet and RNAcompete (RNAC)**

| Figure 4 | a | b | c | d | e | f | g | h |
|---|---|---|---|---|---|---|---|---|
| Gene name | PABPC1 | PCBP2 | PTBP1 | RBFOX1 | SNRPA | SRSF2 | TARDBP | U2AF2 |
| Sampled PDBID | 1cvj | 2py9 | 2adc | 2err | 1aud | 2lec | 4bs2 | 2g4b |
| RNAC ID | 155 | 44 | 269 | 168 | 71 | 72 | 76 | 79 |
| RNAC suggested sequence | ARAAAAM | CCYYCCH | HYUUUYU | WGCAUGM | WUGCACR | GGAGWD | GAAUGD | UUUUUYC |
| NucleicNet suggested sequence | AAAAAAW | WHCYCUWHCYCU | UUUWYU | URHAUGU | AWUGCAH | WNGAGW | RURWAUGA | UUDWWW |
| PDB deposited sequence | AAAAAAA | AACCCUAACCCU | CUCUCU | UGCAUGU | AUUGCAC | UGGAGU | GUGAAUGA | UUUUU |
| Pearson correlation (RNAC PWM Score vs NucleicNet Score) | 0.81 | 0.70 | 0.73 | 0.27 | 0.74 | 0.32 | 0.77 | 0.72 |
| Welch's t-test statistics (highest 10—lowest 10) | 20.7 | 16 | 25.3 | 5.2 | 6.2 | 7 | 1.7 | 20.2 |
| Welch's t-test P-value | 6.10E-13 | 1.90E-09 | 6.70E-13 | 2.40E-04 | 4.90E-05 | 3.90E-06 | 1.10E-01 | 8.30E-09 |

Best matching suggested sequences between RNAC and NucleicNet are underlined. R: A/G, M: A/C, Y: C/T, H: A/C/T, W: A/T, D: A/G/T, and N: A/C/G/U

structural data (accuracy 0.26 as in Fig. 2c inset 2adc). Another example is protein RBFOX1, for which there are only two deposited PDB entries 2err (with RNA sequence UGCAUGU) and 2n82 (with RNA sequence GGCAUGA). Even so, NucleicNet can correctly predict U/A at the first position with a dominant U, which is in agreement with the RNAC suggested sequence (Fig. 4d).

**Preferences in guide strand loading of hAgo2.** As aforementioned, small changes in sequence at the 5′ end of guiding RNA (base 1–8) can lead to variable consequences in RISC assembly and thereafter affect siRNA knockdown efficiency. Therefore, knowing how guide-hAgo2 interaction and loading efficiency are correlated is crucial towards development of efficient RNA-induced silencing tools. To assess NucleicNet's ability in predicting asymmetry in gRNA loading, we compared the NucleicNet score $Q$ with quantitative results from two types of in vivo experiments – immunoprecipitation assay and siRNA knockdown; $Q$ is derived from analysis of a hAgo2 structure (PDBID: 4f3t) and alignment with a trinucleotide conformation library (Supplementary Figs. 4 and 5 and Methods).

**Evaluation on the Ago2-RIP-Seq experiment.** We show that $Q$ can differentiate between guide and passenger sequences from the same precursor miRNA duplexes determined by Ago2 IP followed by small RNA sequencing from different cell lines, namely four from human (acute monocytic leukemia THP-1[26], colon cancer DLD[27], colon cancer HCT116[28], and T cell leukemia[29]) and one from mouse (neuroblastoma N2a[30]). In each dataset, a strand is considered the guide in the duplex when its reads per million (RPM) supersedes its complement by at least 2 orders of magnitude in an Ago2-RIP-Seq experiment (Ago2-RNA Immunoprecipitation and Sequencing)[31]. Duplexes with guide strand having less than 25 RPM are also discarded resulting in a total of 222 duplexes under evaluation (Supplementary Table 4). For each dataset, a histogram of NucleicNet score difference $Q_{guide} - Q_{passenger}$ between the guide and the passenger strands of each duplex is produced (Fig. 5a). A positive difference means that the guide is predicted more favorably than the passenger in binding according to NucleicNet analysis, which is the desired result. In summary, 76% of the tested duplexes show positive differences. To quantify statistical significance of these differences, a paired T-test and a Wilcoxon signed rank test were conducted. Both tests survived $p$-value $< 0.005$ criteria in all datasets confirming NucleicNet's ability in predicting small RNA asymmetry defined from an in vivo setup (Supplementary Note 3, Supplementary Table 3, Supplementary Figs. 7 and 8).

**Evaluation on the siRNA knockdown experiment.** In siRNA knockdown experiments, different guide sequences with different loading efficiency can affect RISC assembly, therefore their silencing efficiency could be different[19,32]. Here we evaluate how well the guide-hAgo2 interactions predicted by NucleicNet can explain these differences. In this regard, we collected knockdown benchmarks for shRNA registered on the Broad Institute RNAi Consortium from the website of a distributor (http://www.sigmaaldrich.com/life-science/functional-genomics-and-rnai.html) and tested for their correlations with the NucleicNet score (data provided in Supplementary Table 5, Supplementary Note 4). To accommodate for heterogeneity in cell lines and target genes, regression analyses were done separately on each entity and were restricted to entities that contain more than one data-points (i.e., different shRNA sequences at base 1–8) (Fig. 5b). Entities with the range of knockdown level narrower than 0.1 were excluded as trends

could not be seen. In summary, 127 data points were used for evaluation, covering 37 genes in total; 90 data points (26 genes) show positive correlations with the NucleicNet score (Fig. 5b), whereas 37 data-points (11 genes) show negative correlation (Supplementary Fig. 9). Although many factors can affect knockdown efficiencies, our results suggest that sequence preferences in guide strand loading is one of them and therefore should be considered in future siRNA designs.

## Discussion

Experimental assays and assay-based computational approaches are quintessential starting points to understand RNA-binding properties of proteins. However, apart from identifying RNA sequence motifs, little can be inferred about the chemistry of base-protein interactions, i.e., the origin of specificity, because atomic and topological details of the RBPs are excluded from analysis. Arguably, this gap of understanding can be filled by elucidating more ribonucleoprotein co-crystals. Nevertheless, even as structural elucidation techniques become more standardized and collections of co-crystals accumulate, efficient ways to exploit this vast abstract structural knowledge have yet to be realized. In this work, by perceiving local physicochemical environment through a deep residual network, we show that meaningful predictions about RNA-binding sites and interaction modes of RNA constituents can be deduced in a pure structure-based computational framework. More importantly, our results show that these learnings on structures can be applied to compare with state-of the-art in vitro and in vivo experimental assay data, suggesting an ability to capture genuine RNA-binding interactions with verifiable biological implications. However, there are few limitations that could shroud structure-based paradigms. First of all, specificity further stabilized by RNA–RNA interactions were not considered; one extreme example is in the ribosomes where the RNA content outnumbers the protein content by folds such that mismatches in RNA–protein interactions may be compensated by RNA–RNA interactions[33]. In our dataset, these proteins are excluded from analysis. Secondly, there are also cases where RNA-protein interaction modes are assisted by base-stacking, base-pairing, and bulges, e.g., in FBF2 and RNase III[17,23]. Even though these parts are distant from the protein surface, they can contribute to enthalpic/entropic cost throughout the binding mechanism, therefore ideally should not be ignored. In the future, structure-based methods may expand to cover training with RNA-structure annotations and RNA-relevant physicochemical features for ribonucleoprotein complexes; this could find utility in understanding target-D/RNA binding in RNA-guided machineries, e.g., Argonautes and CRISPR/Cas. Finally, structure-based methods are ignorant of protein dynamics in RNA-binding mechanisms. For instance, both Argonaute and RNase III would require large conformational changes to incorporate RNAs. In addition, a protein may undergo conformational changes upon binding to different RNA sequences. To this end, sampling of relevant protein conformers may be enhanced by Markov state models[34], normal modes[35], or even large-scale homology modeling[36] if co-crystals are available in protein homologs. Nonetheless, potentials of structure-based methods in recovering chemical binding specificity patterns are very compelling and this genre may become the mainstream in the near future.

## Methods

**Overview of the NucleicNet framework.** In NucleicNet, our goal is to predict on each location of a protein surface, whether the physicochemical environment presented on-site is fit to bind with an RNA and, if affirmative, the most likely type of RNA constituent – Phosphate (P), Ribose (R), Adenine (A), Guanine (G), Cytosine (C), and Uracil (U) – binding to the location (Fig. 1). In the following

subsections, we shall give a summary of our methods. First, we define how relevant non-redundant locations, corresponding to positive and negative examples of RNA-binding sites, are labeled and drawn from the PDB. Then, we describe how physicochemical environments on those locations are perceived by the FEATURE program, which formulates inputs for our deep learning network. Next, we examine the learning strategy and model architecture of NucleicNet to predict the binding class from those physicochemical environments. Finally, we explain how to infer the letter RNA sequences from the NucleicNet predictions.

**Relevant non-redundant locations on protein-RNA complexes**. The surface locations that are 2.5–5.0 Å away from any protein residue are established by three-dimensional coordinates of grid points on a cubic lattice spaced at 1 Å. Relevant locations are selected from the surface locations by considering the topology of the local protein surface and also their bound RNA constituent labels. Non-redundant locations are retrieved by removing grid points associated with homologous proteins from the determined relevant locations. This strict strategy in collecting a relevant non-redundant dataset assures that the training and testing datasets are disjoint under cross validation (Fig. 2) and that the dataset does not carry prior information with respect to proteins.

To determine relevant surface locations, all ribonucleoprotein structures are retrieved from NPIDB[37], an up-to-date server hosting RCSB Protein Data Bank (PDB) structures classified by their bound nucleotides (e.g. RNA–, DNA–, or D/RNA–) (selection criteria of the PDB structures are covered in the SI). To define surface locations (surface grid points) on these PDB structures, Fpocket[38], an alpha-sphere based external program, is adopted to mark out grid points on both buried and solvent-exposed protein surfaces. To provide positive examples of RNA constituent binding sites, geometric centroids of heavy atoms from each constituent are labeled. Surface grid points within 3 Å of these labeled centroids and at most 5 Å away from protein are considered positive relevant locations; each positive relevant location is labeled by a bound RNA constituent. Next, we consider locations where RNA-binding is unlikely. These negative relevant locations are provided by surface grid points selected randomly from space excluded by volumes within 3 Å of any RNA atoms as well as alpha spheres from Fpocket. Note that the number of positive and negative relevant locations are balanced at ratio 2:1 after the removal of redundant locations.

To remove redundant locations, data collected from the PDB are saturated with redundancy. Multiple copies of the same RNA-binding protein chain can exist within the same PDB entry due to the formation of homo- or hetero-multimeric complexes. Homologous chains can also be shared among different PDB entries dedicated to different bound RNA sequences, quality of resolved proteins, and mutants, etc. We define the former situation as internal redundancy and the latter as external redundancy. Often, these homologous chains can share, to large extent, common RNA-binding configurations and physicochemical environments. Using redundancy-inclusive data for training and testing could introduce large bias to the evaluation and overstate the generalizability power of a model. Therefore, redundancy must be removed from the data.

To remove external redundancy, PDB entries are clustered into groups where each entry is linked with another that shares at least one RNA-bound chain with ≥90% BLASTClust sequence homology (Supplementary Figs. 1–3); for each cluster, the PDB entry with the best global resolution is selected. In this way, 483 valid PDB entries becomes 158 cluster and each cluster contributes only one entry to the dataset. In addition, if the selected entry contains multiple copies of the same protein/RNA chains (i.e., internal redundancy), only grid points adhering to the best locally resolved RNAs are retained; grid points adhering to homologous protein chains are also discarded. Local resolution is defined by the average of B-factors on atoms of RNAs; grid points are assigned to adhere the closest RNA/protein residue. Note that the remaining non-redundant grid points are characterized in presence of the internal-redundant protein chains to preserve the intact physicochemical environment. In total, around 280k data points are compiled from the valid PDB entries; two-thirds of which are positive examples. The data points are randomly split into three disjoint folds that disallow both external and internal redundancy, even though members of the same BLAST group (Supplementary Figs. 1–3) can exist within in the same training fold to maximize availability of training data. Note that testing is performed on data points contributed only by the representative member of each BLAST group, where in all three folds, there are 80k such data points.

**Capturing physicochemical environments with FEATURE**. RNA–protein interactions are maintained by physical forces and properties (e.g., electrostatics, hydrophobicity, solvent accessibility, etc.), but the origin and strengths of these interactions are determined by a varied spatial arrangement of chemical components and atoms on the protein surface (e.g., charged residues, hydrogen bond donors/acceptors, etc.). These complicated topological features, which we summarized as physicochemical environments, can be maneuvered into a feature vector, and by leveraging the power of deep learning, to predict RNA-binding partners on protein surface locations – this is the foundation that underlies our NucleicNet method.

In this work, the FEATURE vector framework developed by some of us[12] is adopted to perceive physicochemical environments on three-dimensional protein surfaces. Previously, this framework have been applied to predict cation[39–42] and

ligand/fragment binding sites[43–45]. In those studies, it has been shown as an effective implementation to describe similar binding sites shared by proteins with little structural or sequence resemblance. In contrast to other vector frameworks used by preceding structure[46]-/sequence-based[11] studies, where physical/structural features (at max 60 in total) are accounted in units of residue regardless of their spatial distribution, our physicochemical features are accounted in units of atoms and their discrete radial distribution over a location[12] (Fig. 1 middle panel). As such, these features, 480 in total, preserve a much wider range of details (including atom types, elements, residues, functional groups, secondary structures, charges, hydrophobicity, solvent accessibility, etc. and, their radial distributions) than any other vector framework. For completeness, the list of features under consideration in Halperin et al.[12] is reproduced in Supplementary Table 1; only protein-relevant features are in use, irrelevant features are set to zero. This all-rounded information about physicochemical environments is indispensable for resolving subtle differences among RNA base- and backbone-binding sites (Figs. 2a and 3a–c). It has allowed us not only to tell the spatial region of RNA-binding as in other previous studies, but also to classify these binding sites into six different RNA constituents and deduce specificity towards the RNA bases.

To summarize, after obtaining a set of labeled relevant non-redundant locations, their protein-related physicochemical environment is then characterized under the FEATURE framework in absence of nucleic acid, solvent, substrate, and ions. Hence, each of these locations is annotated by a FEATURE vector and a label that indicates the binding class, and our NucleicNet is trained to predict the label from the FEATURE vector.

**Hierarchical classification of physicochemical environments**. In NucleicNet, our goal is to predict on each location of a protein surface, whether the physico-chemical environment presented on-site is fit to bind with an RNA and, if affirmative, the most likely type of RNA constituent that binds to the location. This is a multi-class classification problem for which end-to-end training is possible, where the seven attainable classes are Phosphate (P), Ribose (R), Adenine (A), Guanine (G), Cytosine (C), Uracil (U), and Non-binding site (X). However, as positive examples of backbone constituents (P and R) are 4–5 times more abundant than that of the nucleobases (A, U, C, G), straightforward deep learning model training suffers from the serious class imbalance problem[47] (Fig. 2a). To alleviate the situation, we therefore adopt a hierarchical classification scheme (Fig. 2a) that balances the data. In the first level, the grid point is classified by a 4-class coarse model, where attainable classes are Base, Ribose, Phosphate, and Non-site, producing a normalized multi-label 4-class score vector. The training of this model requires merging data-points annotated with A/U/C/G to Base. This alleviates the class imbalance problem. To distinguish among the four bases A/U/C/G, a second level classifier is compiled, which does not suffer from the class imbalance problem. A final normalized multi-label 7-class score vector is produced by multiplying the second level outcome (also normalized) with the Base prior from the first level. Consequently, based on such hierarchy, two models are built for the entire problem: one for predicting four coarse classes and the other for distinguishing the four bases. The model architecture common to learners at both levels will be introduced in the next subsection.

**Model architectures**. Architectures of neural networks have been evolving along the development of the deep learning field. From the legendary AlexNet[48] to cutting-edge architectures, such as residual networks (ResNet)[49] and generative adversarial nets (GAN)[50], each of these architectures was designed to push forward the limit of prediction accuracy and resolve specific problems encountered in training on specific categories of data. In this work, considering the complexity of the problem and the convergence rate of the model, ResNet is chosen as our basic unit architecture due to its ability in handling the gradient vanishing problem, which obstructs extensive training of baseline multi-layer convolutional neural network models when deep networks are compiled. Our model is comprised of 16 residual blocks, a fully connected (FC) layer and a final Softmax layer to make a 4-class probability prediction. The residual blocks are considered as the feature extractor and the FC-Softmax is the classifier. In total, 32 convolutional layers are compiled, where each residual block contains 2 convolutional layers. The input tensor from the FEATURE program is of the shape $1 \times 6$ shells $\times 80$ physico-chemical properties. In the convolutional operation, a shared filter of size $1 \times 2 \times 80$ slides across the input, generating an inner product at each position as an intermediate output, which then goes through batch normalization (BN) and element-wise non-linear activation, in our case, rectified linear units (ReLUs)[29], to produce the intermediate output. The use of the BN layer mitigates the internal covariate shift problem[51]. In total, 80 filters are used. Note that to enable a con-sistent size in the output tensor ($1 \times 6 \times 80$), the input tensor is zero-padded. In each residual block, an identical shortcut is added to allow learning of the residual between the input and the second intermediate output. The output from the final residual block is later flattened and fed to a fully connected layer to make four-class probability prediction in the final Softmax layer. All parameters in the network are optimized, with weight decay, under Adam using categorical cross-entropy as the loss function. Training is implemented with TensorFlow. In general, it takes 4 days to train the model at all levels on a Titan X GPU. In Supplementary Discussion, we also compared alternatives to ResNet in the NucleicNet predictor, e.g., shallow machine learning methods and neural networks that do not consider spatial

information. We find that on the grid level prediction, the proposed model, NucleicNet, outperforms all the shallow methods as well as other deep learning architectures under the same experimental setting. Alternative machine learning strategies were also considered, e.g., MAX-AUC[52] and ensemble learning with data sampling[53,54], though issues in run time and overfitting were experienced.

**Obtaining sequence logo with predetermined base locations.** The feed-forward module of NucleicNet annotates each grid point with a normalized score vector that indicates predicted binding probability with respect to the seven attainable classes on that location. For RBPs with predetermined ribonucleoprotein structures (e.g., those compared with the RNAcompete assay in Fig. 4), sequence logo diagrams can be easily generated by considering location $i$ of centroid for the corresponding nucleo-base. As such, the NucleicNet score vectors predicted on grid points within 3 Å of each base centroid are averaged to produce an averaged binding probability $p_i$ (see Supplementary Fig. 4 for the illustrated procedure). Information content $\Xi_i$ on each base position $i$ is then accounted in terms of the following equation, where $p$ is the averaged binding probability on base position $i$ for class $c$:

$$\Xi_i = \log_2 7 + \sum_{c=\{AUCGPRX\}} p_i(c) \log p_i(c) \qquad (1)$$

A sequence logo diagram can then be generated by proportioning the information content $\Xi_i$ according to $P_i(c)$. Class P, R and X, corresponding to Phosphate, Ribose and Non-RNA Binding Sites, are omitted from the logo diagram. Note that a gap is automatically assigned when location $i$ is ≥5 Å away from the protein.

**Scoring RNA letter sequence for Ago2.** Similar to the idea of applying position weight matrix scores (PWM scores) to study DNA sequence-specific binding of transcription-factors[55], the results of NucleicNet for individual protein surfaces can be summarized as an equation $Q$ to score an arbitrary RNA letter sequence input:

$$Q = \max \sum_i^N \log_2 (p_i(b) T_{i,i+1}). \qquad (2)$$

This equation, which we refer to as a fixed hidden Markov model (HMM), is comprised of an emission probability $p_i$ and a transition probability $T_{i,i+1}$. Our goal is to assimilate NucleicNet outputs via $p_i$ and $T_{i,i+1}$ and to consider geometric constraints put forward by the covalent bond network and the torsional space of genuine RNA strands. The hidden states are locations indexed by $i$ of bases relevant to a continuous RNA strand bound to the RBP with the letter sequence of length $N$. The emission probability $p_i(b)$, referring to the binding probability of base $b$ on the RNA sequence, is obtained by averaging the NucleicNet output within 3 Å of the base location $i$ (see Supplementary Figs. 4 and 5 for illustrated process). Note that $p_i(b)$ here is normalized among the bases. The transition probability $T_{i,i+1}$ refers to the transition probability between bases $i$ and $i + 1$ on a continuous RNA strand from 5′ to 3′ end. In case, base locations are predetermined by ribonucleoprotein co-crystals, transitions between consecutive bases as well as their locations $i$ are certain, then $T_{i,i+1} = 1$ and $p_i(b)$ can be deduced by averaging the NucleicNet output on locations just as we generate logo diagrams. The equation $Q$ is then reduced to an ordinary PWM scoring function; RNA string sequences of length $N$ are then evaluated by sliding across the co-crystal-native RNA strand locations to obtain a maximum in $Q$, which is implemented to calculate the NucleicNet score for comparison with the RNAC score (Fig. 4a–h, Supplementary Note 2).

We also investigate the situation where base locations referring to a continuous RNA strand are unknown but NucleicNet predicted RNA-binding sites are clearly directed by a phosphate-ribose backbone (e.g., in RNA-guided situations, for instance, Ago2 in Fig. 5, Supplementary Fig. 4). In this case, score $Q$ cannot be easily generated as in the case of co-crystals because those hidden locations and their transition probabilities $T_{i,i+1}$ are unknown, even though $p_i(b)$ can still be calculated from the NucleicNet outputs around location $i$ once locations are approximated. In the next section, we outline how these unknowns can be efficiently estimated by aligning top predicted binding sites of RNA constituents with a conformational library of RNA trinucleotides. In this case, the score $Q$ can then be obtained by maximizing over all possible $i$, $i + 1$ transition paths, when an RNA letter sequence of length $N$ is enquired.

A continuous RNA strand may be considered as a transition graph between locations of consecutive bases, where base identities can be expressed by an emission probability $p_i(b)$ on each node indexed by a location $i$ referring to the location of a base $b$ on an RNA strand bound to an RBP. In case, where these locations are hidden, the transition probability $T_{i,i+1}$ is unknown. However, these transitions are certainly constrained, irrespective of the strand length, by the covalent bonds and the torsional space of the RNA[56,57]. Therefore, they can be estimated by screening a database of RNA geometries that are tolerated by series of predicted RNA-binding sites on the RBP surface. In particular, for cases where predicted RNA-binding sites are clearly directed by a phosphate-ribose backbone (e.g., in Ago2 where the RBP is known to work in an RNA-guided manner), this trail of backbone-binding sites and intermittent base binding sites are visually indicative for a continuous RNA strand (Supplementary Fig. 4). To efficiently screen out binding sites relevant to a continuous RNA strand in this case, top 10% of binding sites reported by NucleicNet are aligned with a non-redundant library of trinucleotide conformations adopted from Humphris-Narayanan et al.[56]. This

library was compiled from ribonucleoprotein complexes in the PDB by binning over the pseudo-torsional space of RNA backbones[56], from which, the 15°-bin library containing 296 conformers is chosen for our purpose. To compile a comprehensive trinucleotide conformer library, the 15°-bin library is permuted to cover all 4³ possible trinucleotide sequences in atomic details for each conformer; the resultant 18944 trinucleotide conformers are optimized briefly under a AMBER99SB-ILDN force field[58] to assure proper geometry. Finally, these trinucleotides are reduced to centroids of their RNA constituents (nodes) resulting in some 9-nodes coarse-grained models ready to be aligned with the top binding sites. The clique-alignment process is done with a Bron-Kerbosch algorithm[59], where only ≥7-cliques that show no atomic clash with the protein are retained. The 7-clique is chosen such that transition between consecutive $i$, $i + 1$ bases (i.e., 2 Base nodes on the 9-node model) must be guided by at least five backbone constituents. These criteria assure that the proposed binding sites are geometrically feasible. To systematically assess how these aligned 3-mers can contribute to a continuous strand, we formulate the problem as a fixed HMM. Hypothetical base locations are the hidden states. To propose these locations, the Euclidean space covered by the aligned Base nodes is partitioned into multiple Voronoi cells seeded by k-means centers. To express the identity of the base, these Voronoi cells, each represent a hypothetical base location, are characterized by emission probabilities $p_i$ averaged from grid points within 3 Å of a k-means center. Then, transitions, regarding consecutive bases within the same aligned clique, between different Voronoi cells are counted and symmetrized as an estimate of transition probability $T_{i,i+1}$. In case of Ago2, since it is ascertained that the 5′ location is situated in the Mid domain[20], a certain starting probability of one is assigned to a cell located in the Mid domain that is furthest away from any other cells. The 5′ to 3′ direction of transition is then ascertained by the ranking distance to this starting Voronoi cell; direction of the edge on the transition graph allows only transition from a high rank to a low one. Details of the HMM are presented in Supplementary Figs. 5 and 6. With $p_i$ and $T_{i,i+1}$ affixed, score $Q$ can then be calculated using the equation presented above.

**Reporting summary.** Further information on research design is available in the Nature Research Reporting Summary linked to this article.

## Data availability
The data that support the findings of this study are available from the corresponding author upon reasonable request. The authors declare that all other data supporting the findings of this study are available within the paper and its supplementary information files.

## Code availability
NucleicNet is hosted on our webserver http://www.cbrc.kaust.edu.sa/NucleicNet/. The source code for a working version of NucleicNet is available at https://github.com/NucleicNet/NucleicNet.

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

## Acknowledgements

We are grateful to Wei Wang for helpful discussions. Figure 1 was created by Heno Hwang, scientific illustrator at King Abdullah University of Science and Technology (KAUST). This work was supported by grants from KAUST to X.G. (BAS/1/1624-01, FCC/1/1976-18-01, FCC/1/1976-23-01, FCC/1/1976-25-01, FCC/1/1976-26-01, and FCS/1/4102-02-01) and funding from the KAUST to X.G. and X.H. (URF/1/3007-01). The Hong Kong Research Grant Council (HKUST C6009-15G, AoE/M-09/12, and AoE/P-705/16) and Innovation and Technology Commission (ITCPD/17-9 and ITC-CNERC14SC01) to X.H.; L.F., Y.F.L., and W.C. were supported by Research Grant from Science and Technology Innovation Commission of Shenzhen Municipal Government (No. KQTD20180411143432337 and JCYJ20170307105752508). Part of bioinformatics analysis was supported by the Center for Computational Science and Engineering of Southern University of Science and Technology.

## Author contributions

X.G., X.H. and R.B.A. conceived this study. H.J., A.H. and T.L. initiated the study. J.H.M.L extracted datasets from PDB. Y.L. and J.H.M.L. implemented the deep learning model. J.H.M.L. and F.K.S. designed the scoring interface for letter sequences. Y.K.L., Y.F.L. and L.F. collected experimental data concerning hAgo2; W.C., X.G., and J.H.M.L. designed statistical tests for these data. R.U., J.H.M.L. and Y.L. designed the webserver. J.H.M.L, L.Z. and Y.L. wrote the manuscript under supervision of X.G., X.H., L.Z. and W.C. All authors are involved in discussion and finalization of the manuscript.

## Competing interests

The authors declare no competing interests.
