## [Peer Review File · Nature Communications]

Reviewers' comments:

Reviewer #1 (Remarks to the Author):

Protein-RNA interaction detection and analysis is an important field in post-transcriptional regulation researches. In this paper, by integrating local physicochemical characteristics of the protein structure surface, a deep learning framework, NucleicNet, is proposed to predict the binding preference of different RNA constituents on protein surface. It's an interesting work in this field, and the method and evaluation approach seems sound. This study may help us to explore protein-RNA interactions in more detail. My concerns regarding the presented work are as follows:

1. The carefully selected and constructed data sets are employed for training and comparison. Does these specific training data and repeated cross validation process lead to over-fitting? For example, in section "Complex Spatial Patterns of RNA-binding Sites Reproduced by NucleicNet", the results and discussion on three exemplary RBPs show the effectiveness of NucleicNet. Besides these three specific RBPs, could this method achieve be equally effective on other types of RBPs? Whether the accuracy of results are influenced by the specific training set?
2. The detailed process diagram of pipeline is needed to help to understand the detailed functions of the whole framework.
3. The whole framework provide several prediction and analysis modules, however, besides the collection of different features and the using of deep learning technique, what are the theoretical and technical innovations in Information Science compared with the existing methods? It seems the good performance are due to the additional prior and guidance information. If these information are employed by other methods, can they achieve the same or even better results?
4. Is it reasonable to manually set the thresholds of all features? What are the descriptors of these features?
5. It is a very basic hierarchical classification strategy employed by NucleicNet. If other better classification methods are introduced to deal with imbalance problems, will the prediction effect be better?

Reviewer #2 (Remarks to the Author):

In this paper, the authors proposed a novel computational method, in fact, first-of-its-kind, to identify potential binding sites and binding specificities of RNA constituents on any surface location on any given three-dimensional protein structure. In particular, they developed a deep learning approach based on a structural framework which allows nucleotide specificities to be identified by local characteristics and subsequently visualized. This technique also allowed them to construct easy-to-read logo diagrams when the binding pocket of the RNA chain is known on the protein surface, or allowed them to predict the binding pocket as well as the preferred RNA sequence for unknown RNA binding proteins. Compared to previous structural methods which only make binary predictions on whether sites on proteins are RNA-binding or not, the 7-class prediction presented by this current study is significantly more challenging, sound, and useful. Comprehensive experiments on cross-fold validation on protein-RNA complexes in PDB, in vitro assay, and in vivo assay demonstrated the impressive performance of the proposed method. They also made their method available to the public through a webserver. Overall, I find this a very interesting work and a very useful structural bioinformatics tool that will be of broad interest to structural biology, bioinformatics, and biochemistry communities, which can make significant impact in the field. The paper is beautifully written, with a clear logic flow, careful explanations, and comprehensive and convincing experiments. I enjoyed reading it. I have the following comments/questions for the authors to address:

1. The authors used deep learning and FEATURE framework to achieve great performance in predicting the binding preference of RNA on the protein surface. I found that the features used in this manuscript are not exactly the same as the original FEATURE framework. Please clarify the selection procedure of the used features.

2. In the Online Methods section, the authors gave detailed descriptions of the model and training procedure, which should be sufficient for reproduction. Please also report the running time of the testing procedure. Such information would be useful to the users.
3. In the Online Methods section, the authors provided descriptions about their deep learning architecture. I suggest the authors give the simplified information in the legend of Figure 1, so that the readers would be able to understand the deep learning model without going into the details in the latter part of the paper.
4. In Figure 2(a), we can clearly see that the data is imbalanced. Please explain how you deal with the imbalanced data.
5. In Figure 2(b), MCC is not defined.
6. In Table 1, the macro-average and micro-average are used to evaluate the imbalanced multi-class classification results. What is their main difference? Please give formal definitions of those terms in SI or the Online Methods section, and refer them in the main text.
7. Deep learning is known to have a tendency to overfitting. Although the large data size here can greatly alleviate the issue, additional techniques should be applied to further prevent the issue. Please clarify the techniques utilized to deal with the overfitting problem.
8. In reality, some RNA-protein interactions can occur in the interface between two or multiple proteins. Please clarify if your method can deal with such a situation, and if so, how?

We are very grateful to the two reviewers for their thoughtful and thorough comments, which definitely helped us improve our paper greatly. We have revised the paper following all of their comments. Below please find the point-by-point response to all the reviewers' comments.

=====

Reviewer #1 (Remarks to the Author):

1. The carefully selected and constructed data sets are employed for training and comparison. Does these specific training data and repeated cross validation process lead to over-fitting? For example, in section "Complex Spatial Patterns of RNA-binding Sites Reproduced by NucleicNet", the results and discussion on three exemplary RBPs show the effectiveness of NucleicNet. Besides these three specific RBPs, could this method achieve be equally effective on other types of RBPs? Whether the accuracy of results are influenced by the specific training set?

Answer: Thank you very much for the excellent question. We completely agree that overfitting is a common issue in many machine learning works and it is difficult to avoid overfitting. In this paper, we made a number of efforts to avoid overfitting as much as possible:

- 1) A carefully-curated, comprehensive, non-redundant dataset of protein-RNA complex structure dataset was formed. We carefully selected and curated non-redundant dataset from all protein-RNA complex structures from PDB (see Methods), which consists of 158 complex structures, resulting in about 280,000 grid points in the dataset.
- 2) Three-fold cross-validation was performed over this dataset containing 158 complex structures. Each time, two folds were used for training and one for testing. Between folds, BLASTClust sequence homology of $\geq 90\%$ was disallowed (see Methods), making sure that training and testing has no similar structures. Notice that the granularity of this cross-validation is individual proteins, instead of grid points, which eliminates bias in the size of proteins.
- 3) We performed multiple rounds of 3-fold cross validation, with different random split, and the statistical results are stable. Thus the accuracy of the results is not sensitive to the specific training set.
- 4) During deep learning training, we applied different techniques to avoid overfitting, including residual networks, dropout, weight decay, batch normalization, etc. In addition, we monitored the performance on the validation dataset (a part of the training dataset, without any overlap with the testing dataset) and stopped the training when the performance degraded.
- 5) We conducted comprehensive validation on *in vitro* data (on all the RBPs for which both RNACompete data and PDB structures are available) and *in vivo* data (Ago2-RIP-Seq experiment in five cell lines and siRNA knockdown experiment in four cell lines).

Therefore, although we cannot theoretically guarantee that the overfitting issue has been eliminated, we tried our best to reduce the overfitting risk to the lowest level. The distribution of the macro-average precision was given in Figure 2(c) (copied below), where proteins covered in case studies are marked out with their PDBID on the inset line diagram to indicate their performance, which shows that the accuracy of the case studies spreads over a wide range, instead of cherry-picking.

In addition to the statistical evaluation, the three exemplary RBPs were chosen under the following requirements to give readers a comprehensive feeling of the power of the model: 1) one with direct recognition of RNA motifs (FBF2), one with indirect RNA-guided manner (hAgo2), and one with double-strand RNA (Aa-RNase III); 2) the detailed binding preference of different bases at important locations is known to biologists, because the protein-RNA complex structure only gives one snapshot of the many possible interactions, which does not provide the preference of different bases and cannot be used directly to compare with the prediction of our method. Therefore, the choice of such exemplary RBPs is quite limited. We mainly used the three examples to demonstrate the output of our method, its performance for different types of interactions, and the visualization of the results. For comprehensive and statistical evaluation, we refer the readers to 3-fold cross validation, *in vitro* and *in vivo* experiments.

2. The detailed process diagram of pipeline is needed to help to understand the detailed functions of the whole framework.

Answer: Thank you for the thoughtful suggestion. We have now supplied a pipeline in the form of a text flow chart for each of the three main functions of our program. Please kindly refer to SI Section 1.4 (SI Figure 4). We also include it below for your reference:

To summarize, “First, users are required to provide atomic coordinates of a protein structure (e.g., a .pdb file). A common prediction protocol then follows. (SI Fig. 4a) Each location on the protein surface is marked a point such that the entire surface of the protein is seeded by a layer of evenly distributed grid points. Subsequently, local physicochemical features of each location are analysed to produce an input vector for NucleicNet, our trained deep learning module. Each location is then scored for labels indicating binding preference of the 7 classes. These score vectors can be used by three different utility modules depending on the situation. (SI Fig. 4b) If the RNA-protein structure were predetermined, i.e., the binding sites are known, our Sequence Logo module can be applied to predict binding specificity on each nucleoside position as logo diagrams. If RNA-protein structure were not predetermined, i.e., the binding sites are unknown, Hidden Markov Model (HMM) needs to be constructed in order to ascertain positions of nucleosides and the resultant HMM can be used as a scoring interface for letter sequences. The Visualization module applies to both situations.”

3. The whole framework provide several prediction and analysis modules, however, besides the collection of different features and the using of deep learning technique, what are the theoretical and technical innovations in Information Science compared with the existing methods? It seems the good performance are due to the additional prior and guidance information. If these information are employed by other methods, can they achieve the same or even better results?

Answer: Thank you very much for raising this question up. Yes, although our main contribution is to propose the first method that can quantitatively predict the binding preference of different RNA constituents on any surface location of any protein structure, we completely agree with you that the proposed method should be compared with other traditional machine learning algorithms or deep learning models with different architectures.

Following your comment, we have compared NucleicNet, on the same dataset, with a number of representative machine learning methods, including Support Vector Machine (SVM), Random Forest (RF) and K-Nearest Neighbor (KNN), as well as other deep learning models of different architectures (e.g. 'FCNet', a model composed of multiple fully-connected neural network layers and 'CNN_1D', a model which convolves across the whole feature vector without considering the spatial information). The three-fold cross-validation results are shown in the tables below (these tables have also been added into SI as SI Table 6). We can find that on the grid level prediction, the proposed model, NucleicNet, outperforms all the other methods under the same experimental setting.

Level 0 Prediction							
Method	Accuracy	Kappa	Macro-precision	Macro-recall	Micro-F1	Macro-F1	AUROC
NucleicNet	0.726	0.626	0.699	0.700	0.726	0.692	0.912
FCNet	0.705	0.589	0.679	0.662	0.705	0.664	0.905
CNN_1D	0.709	0.600	0.675	0.676	0.709	0.672	0.903
SVM	0.661	0.531	0.617	0.617	0.661	0.613	0.848
RF	0.695	0.576	0.668	0.615	0.695	0.568	0.892
KNN	0.649	0.501	0.663	0.598	0.649	0.603	0.790

Level 1 Prediction							
Method	Accuracy	Kappa	Macro-precision	Macro-recall	Micro-F1	Macro-F1	AUROC
NucleicNet	0.445	0.225	0.427	0.415	0.445	0.404	0.673
FCNet	0.424	0.187	0.403	0.384	0.424	0.379	0.645
CNN_1D	0.396	0.157	0.355	0.354	0.396	0.348	0.629
SVM	0.366	0.119	0.330	0.330	0.366	0.323	0.586
RF	0.381	0.076	0.368	0.298	0.381	0.242	0.614
KNN	0.438	0.228	0.419	0.413	0.438	0.402	0.629

The consistent outperformance of NucleicNet implies that other than the important information that our features capture, our deep learning model itself has advantages on making use of these features over the other methods on solving this problem. In particular, the choice of the residual blocks and the design of the convolution filters fits this problem the best. In addition, there are important theoretical and technical innovations that allow us to tackle the problem in a new depth. First, spatial information is engineered into our input by considering features in layers of shells and the information convolves across shells, where some of these features (e.g. residue-specific features, atomic attributes) as well as the spatial information were not considered by previous methods. Second, researchers in this field were more interested in understanding RNA-protein interaction through the perspectives of sequence analysis; spatial analysis on the atomic structure of proteins was still in the early stage of development and researchers had to reduce the problem to much simpler ones, i.e., binary prediction. However, given the surge in computing power and availability of three-dimensional atomic information, we made this first attempt to explore the solution on the grid point level around the protein surface, which were not studied before. Further, to introduce our predictions as scores for sequences, we have integrated predictions on

the grid point level by stacking a specific probabilistic graphic model, hidden Markov model (HMM), on top of our deep learning model. Although a more interesting approach could be to pre-train the HMM model with external sequence data and to fine tune both the deep learning model and the HMM in an end-to-end fashion, our method can be considered as a first workable pipeline with satisfactory performance.

4. Is it reasonable to manually set the thresholds of all features? What are the descriptors of these features?

Answer: Thank you very much for the comment. In SI Section 1.2 (SI Table 1), we have supplied a full list of descriptors for the features in use; a detailed reference to each of these descriptors can be found in reference 12 “Halperin, I., Glazer, D. S., Wu, S. & Altman, R. B. The FEATURE framework for protein function annotation: modeling new functions, improving performance, and extending to novel applications. BMC Genomics 9, S2 (2008)”.

In our case, a fixed list of features are in use; some of these features were constantly set to 0.0 and there is no other manually set thresholds in the features. As the feature vector obtained from the FEATURE program can contain features that are specific to crystal structures (e.g. number of crystallographic water RESIDUE_NAME_IS_HOH), to avoid capturing these unnecessary information, they are all set to 0.0 in the input vector.

There are 8 such “unused” features including “RESIDUE_NAME_IS_HOH” (related to presence of crystallographic water), “RESIDUE_NAME_IS_OTHER” (related to unnatural or modified amino acids), “CLASS1_IS_UNKNOWN” (related to unnatural or modified amino acids not recognized by the FEATURE program), “CLASS2_IS_UNKNOWN” (related to unnatural amino acids), “MOBILITY” (related to B-factor), “ATOM-NAME-IS-OTHER” (related to presence of ion/ligand), “ATOM-TYPE-IS-Na” (related to presence of sodium), “ATOM-TYPE-IS-Ca” (related to presence of calcium).

5. It is a very basic hierarchical classification strategy employed by NucleicNet. If other better classification methods are introduced to deal with imbalance problems, will the prediction effect be better?

Answer: Thank you very much for this excellent question. The data imbalance issue is indeed a factor that limits the model’s performance and it is worthy to have a more thorough analysis and discussion. Following your suggestion, we first conducted experiments to show the necessity of a hierarchical classification strategy. If we train a NucleicNet to predict the 7 class directly without considering the hierarchical structure of the labeling space, the performance is shown in the table below (now supplied in the SI as SI Table 7).

Metrics	NonSite	Phosphate	Ribose	Adenine	Guanine	Uracil	Cytosine
F1-score (macro) w/ Hierarchy	0.90	0.70	0.63	0.47	0.38	0.48	0.32

F1-score (macro) w/o Hierarchy	0.88	0.63	0.55	0.17	0.04	0.13	0.00
Recall (macro) w/ Hierarchy	0.88	0.82	0.63	0.46	0.38	0.45	0.37
Recall (macro) w/o Hierarchy	0.90	0.69	0.53	0.24	0.03	0.12	0.00
Precision (macro) w/ Hierarchy	0.92	0.61	0.63	0.48	0.38	0.51	0.29
Precision (macro) w/o Hierarchy	0.87	0.58	0.57	0.13	0.07	0.14	0.09

As shown in the table, the model can have very good performance on those large classes, like ‘NonSite’, ‘Phosphate’ and ‘Ribose’. However, on small but biologically important classes, such as ‘Guanine’ and ‘Cytosine’, the model’s performance is unacceptable. On the other hand, with hierarchical classification, NucleicNet’s performance on those small classes has been largely improved. Furthermore, we also applied some other techniques, such as weighted loss, to handle the data imbalance problem.

We next explored other more advanced machine learning methods to further push the model’s performance. We tried two additional methods, MAX-AUC¹ and ensemble learning² with data sampling³, under our current framework. The MAX-AUC method can directly maximize the empirical area under the ROC curve (AUC), which is an unbiased measurement for imbalanced data.¹ As for ensemble learning with data sampling, we combined the prediction power of a 20-member ensemble of NucleicNet, with the help of data sampling to further cope with the data imbalance issue. Their performance is shown in the following tables (they have been added into SI as SI Table 6). The first table is for level 0 prediction and the second table is for level 1 prediction. ‘NucleicNet_auc’ represents the combination of MAX-AUC and NucleicNet. ‘NucleicNet_en’ represents the ensemble version of NucleicNet with data sampling.

Level 0 Prediction							
Method	Accuracy	Kappa	Macro-precision	Macro-recall	Micro-F1	Macro-F1	AUROC
NucleicNet_en	0.766	0.681	0.736	0.730	0.766	0.718	0.922
NucleicNet	0.726	0.626	0.699	0.700	0.726	0.692	0.912
NucleicNet_auc	0.711	0.602	0.703	0.671	0.711	0.670	0.902

Level 1 Prediction							
Method	Accuracy	Kappa	Macro-precision	Macro-recall	Micro-F1	Macro-F1	AUROC
NucleicNet_en	0.451	0.232	0.424	0.411	0.451	0.408	0.696
NucleicNet	0.445	0.225	0.427	0.415	0.445	0.404	0.673
NucleicNet_auc	0.409	0.171	0.377	0.364	0.409	0.354	0.632

As shown in the above tables, all three methods can achieve reasonable performance. Compared to the original version of NucleicNet, 'NucleicNet_auc' encountered overfitting problem, so the latter did not surpass the former in performance. In 'NucleicNet_en', significantly more computational resource and running time has been paid for the performance gain, especially on level 0. However, from the perspectives of users, the original version of NucleicNet, which has the best single-model performance, should be of their most practical interest, and is thus set as the default option in our program.

Reference

1. Wang, S., Sun, S. and Xu, J., 2016, September. AUC-Maximized deep convolutional neural fields for protein sequence labeling. In *Joint European Conference on Machine Learning and Knowledge Discovery in Databases* (pp. 1-16). Springer, Cham.
2. Schmidhuber, J., 2015. Deep learning in neural networks: An overview. *Neural networks*, 61, pp.85-117.
3. Van Hulse, J., Khoshgoftaar, T.M. and Napolitano, A., 2007, June. Experimental perspectives on learning from imbalanced data. In *Proceedings of the 24th international conference on Machine learning* (pp. 935-942). ACM.

=====

Reviewer #2 (Remarks to the Author):

1. The authors used deep learning and FEATURE framework to achieve great performance in predicting the binding preference of RNA on the protein surface. I found that the features used in this manuscript are not exactly the same as the original FEATURE framework. Please clarify the selection procedure of the used features.

Answer: Thank you for bringing this up. In SI Section 1.2 (SI Table 1), we have supplied a full list of features in use; a detailed reference to these features can also be found in reference 12 "Halperin, I., Glazer, D. S., Wu, S. & Altman, R. B. The FEATURE framework for protein function annotation: modeling new functions, improving performance, and extending to novel applications. *BMC Genomics* 9, S2 (2008)".

The features in use as input for NucleicNet are not exactly the same as the original FEATURE framework. Specifically, some of the original features are not in use in NucleicNet; they are constantly set to 0.0 in the input fed to NucleicNet. The reason we have rejected the use of these features is that they can contain features that are specific to crystal structure (e.g. number of crystallographic water RESIDUE_NAME_IS_HOH).

There are 8 such “unused” features including “RESIDUE_NAME_IS_HOH” (related to crystallographic water), “RESIDUE_NAME_IS_OTHER” (related to unnatural or modified amino acids), “CLASS1_IS_UNKNOWN” (related to unnatural or modified amino acids not recognized by the FEATURE program), “CLASS2_IS_UNKNOWN” (related to unnatural amino acids), “MOBILITY” (related to B-factor), “ATOM-NAME-IS-OTHER” (related to presence of ion/ligand), “ATOM-TYPE-IS-Na” (related to presence of sodium), “ATOM-TYPE-IS-Ca” (related to presence of calcium).

2. In the Online Methods section, the authors gave detailed descriptions of the model and training procedure, which should be sufficient for reproduction. Please also report the running time of the testing procedure. Such information would be useful to the users.

Answer: Thank you for the comment, with which we fully agree. The running time is an important piece of information for the users. With the help of GPUs, the deep learning part of our method is very fast, which can process one protein in 10 seconds. Although the whole pipeline of NucleicNet contains some time-consuming pre-processing and post-processing steps, we can process one protein in one hour. Following your suggestion, we have added the information in SI Section 4.

3. In the Online Methods section, the authors provided descriptions about their deep learning architecture. I suggest the authors give the simplified information in the legend of Figure 1, so that the readers would be able to understand the deep learning model without going into the details in the latter part of the paper.

Answer: Thank you for the thoughtful suggestion. By making the legend of the main figure clearer, the manuscript will indeed be more readable. Following your suggestion, we have added the following description in the legend of Figure 1.

“The FEATURE vectors are reshaped based on the shell information before fed into the neural network. When performing convolution, the filter will convolve across different shells, taking the spatial information into consideration. In addition to that, we used residual networks to make the convolutional layers learn the residual between the input and the output, which allows us to stack more layers and thus extract ‘deep’ features from the inputs.”

4. In Figure 2(a), we can clearly see that the data is imbalanced. Please explain how you deal with the imbalanced data.

Answer: Thank you very much for the excellent comment. As we replied to comment #5 of Reviewer 1, we made the following efforts in the original version of NucleicNet. First of all, we took advantage of the

hierarchical structure of the labeling space and trained one model for each level. By doing that, we were able to obtain reasonable performance on the small while important classes. Secondly, we further used weighted loss to emphasize those small classes.

But as you and Reviewer 1 pointed out, the class imbalance problem is so important that we further dealt with it in this revision. To sum up, we tried two other techniques. Firstly, we tried to directly optimize AUROC, which is an unbiased measurement for imbalanced data. Secondly, we tried ensemble learning with data sampling. The second method has further improved the model performance.

In particular, we tried two additional methods, MAX-AUC¹ and ensemble learning with data sampling, under our current framework. The MAX-AUC method can directly maximize the empirical area under the ROC curve (AUC), which is an unbiased measurement for imbalanced data. As for ensemble learning with data sampling, we combined the prediction power of a 20-member ensemble of NucleicNet, with the help of data sampling to further cope with the data imbalance issue. Their performance is shown in the following tables (they have been added into SI as SI Table 6). The first table is for level 0 prediction and the second table is for level 1 prediction. ‘NucleicNet_auc’ represents the combination of MAX-AUC and NucleicNet. ‘NucleicNet_en’ represents the ensemble version of NucleicNet with data sampling.

Level 0 Prediction							
Method	Accuracy	Kappa	Macro-precision	Macro-recall	Micro-F1	Macro-F1	AUROC
NucleicNet_en	0.766	0.681	0.736	0.730	0.766	0.718	0.922
NucleicNet	0.726	0.626	0.699	0.700	0.726	0.692	0.912
NucleicNet_auc	0.711	0.602	0.703	0.671	0.711	0.670	0.902

Level 1 Prediction							
Method	Accuracy	Kappa	Macro-precision	Macro-recall	Micro-F1	Macro-F1	AUROC
NucleicNet_en	0.451	0.232	0.424	0.411	0.451	0.408	0.696
NucleicNet	0.445	0.225	0.427	0.415	0.445	0.404	0.673
NucleicNet_auc	0.409	0.171	0.377	0.364	0.409	0.354	0.632

As shown in the above tables, all three methods can all achieve reasonable performance. Compared to the original version of NucleicNet, ‘NucleicNet_auc’ encountered overfitting problem, so the latter did not surpass the former in performance. In ‘NucleicNet_en’, significantly more computational resource and running time has been paid for the performance gain, especially on level 0. However, from the perspectives of users, the original version of NucleicNet, which has the best single-model performance, should be of their most practical interest, and is thus set as the default option in our program.

5. In Figure 2(b), MCC is not defined.

Answer: Thank you very much for pointing this out. 'MCC' represents Matthews correlation coefficient. It is defined as

$$MCC = \frac{TP * TN - FP * FN}{\sqrt{(TP + FP) * (TP + FN) * (TN + FP) * (TN + FN)}}$$

where TP, TN, FP, FN are true positives, true negatives, false positives and false negatives. This formal definition has been added to SI Section 2.1 and referred in the main text.

6. In Table 1, the macro-average and micro-average are used to evaluate the imbalanced multi-class classification results. What is their main difference? Please give formal definitions of those terms in SI or the Online Methods section, and refer them in the main text.

Answer: Thank you for the question. Macro-average and micro-average are two slightly different ways of summarizing the performance information of different classes in multi-class classification problems. Both are important when we have imbalanced data. In brief, as for macro-average, we calculated the base metrics for each label and then we performed an unweighted averaging to get the macro-average. Basically, it treated each class equally without considering the class size. On the other hand, micro-average performed averaging on the sample level by counting the true positives, true negatives, false positives and false negatives of each class, and then calculating the metrics accordingly. Essentially, the larger a class is, the larger influence will the class have on the micro-average.

Formally, take macro-precision and micro-precision as an example,

$$Precision_i = \frac{TP_i}{TP_i + FP_i'}$$

$$Macro - precision = \frac{\sum_{i=1}^k Precision_i}{k},$$

$$Micro - precision = \frac{\sum_{i=1}^k TP_i}{\sum_{i=1}^k TP_i + \sum_{i=1}^k FP_i'}$$

where k is the number of classes; TP_i is true positives of the class i ; FP_i' is the false positives of the class i . As you suggested, we have added it in the SI Section 2.1.

7. Deep learning is known to have a tendency to overfitting. Although the large data size here can greatly alleviate the issue, additional techniques should be applied to further prevent the issue. Please clarify the techniques utilized to deal with the overfitting problem.

Answer: Thank you for the excellent point. As we replied to comment #1 of Reviewer 1, we completely agree that overfitting is a common issue in many machine learning works and it is difficult to avoid overfitting. In this paper, we made a number of efforts to avoid overfitting as much as possible:

- 1) A carefully-curated, comprehensive, non-redundant dataset of protein-RNA complex structure dataset was formed. We carefully selected and curated non-redundant dataset from all protein-RNA complex structures from PDB (see Methods), which consists of 158 complex structures, resulting in about 280,000 grid points in the dataset.
- 2) Three-fold cross-validation was performed over this dataset containing 158 complex structures. Each time, two folds were used for training and one for testing. Between folds, BLASTClust sequence homology of $\geq 90\%$ was disallowed (see Methods), making sure that training and testing has no similar structures. Notice that the granularity of this cross-validation is individual proteins, instead of grid points, which eliminates bias in the size of proteins.
- 3) We performed multiple rounds of 3-fold cross validation, with different random split, and the statistical results are stable. Thus the accuracy of the results is not sensitive to the specific training set.
- 4) During deep learning training, we applied different techniques to avoid overfitting, including residual networks, dropout, weight decay, batch normalization, etc. In addition, we monitored the performance on the validation dataset (a part of the training dataset, without any overlap with the testing dataset) and stopped the training when the performance degraded.
- 5) We conducted comprehensive validation on *in vitro* data (on all the RBPs for which both RNACompete data and PDB structures are available) and *in vivo* data (Ago2-RIP-Seq experiment in five cell lines and siRNA knockdown experiment in four cell lines).

Therefore, although we cannot theoretically guarantee that the overfitting issue has been eliminated, we tried our best to reduce the overfitting risk to the lowest level.

8. In reality, some RNA-protein interactions can occur in the interface between two or multiple proteins. Please clarify if your method can deal with such a situation, and if so, how?

Answer: Thank you very much for raising this point. In our training dataset, oligomeric protein assemblies that interact with RNAs were included. Their protein surface features were accounted by applying the FEATURE program in respective grid points without removing any of the protein elements/chains. In particular, when we built up the non-redundant dataset, to make sure interface features in the oligomeric proteins were properly captured by the FEATURE program, we only removed grid points associated with the homologous proteins; none of the atoms from proteins were removed. Therefore, features at the interface were preserved in the training dataset; even though we did not design features specific to protein-protein interfaces.

In predictions, we assumed that the input protein structure is properly assembled. As long as large conformational change is not involved (as discussed on p. 2-3 of Online Method, See subsection "Removing redundant locations."), the features fed into the network are valid with surface features at interface preserved, if any.

REVIEWERS' COMMENTS:

Reviewer #1 (Remarks to the Author):

The authors have fixed all my comments, and I suggest to accept this manuscript.

Reviewer #2 (Remarks to the Author):

The authors have addressed all my comments.